# Comparative Analysis of miRNA Expression Profiles in Skeletal Muscle of Bian Chickens at Different Embryonic Ages

**DOI:** 10.3390/ani12081003

**Published:** 2022-04-13

**Authors:** Kai-Zhi Zhou, Peng-Fei Wu, Xin-Chao Zhang, Xuan-Ze Ling, Jin Zhang, Li Zhang, Pei-Feng Li, Tao Zhang, Qing-Yu Wei, Gen-Xi Zhang

**Affiliations:** 1College of Animal Science and Technology, Yangzhou University, Yangzhou 225000, China; zkz282816@163.com (K.-Z.Z.); wu_p_fei@163.com (P.-F.W.); zpf_xc@163.com (X.-C.Z.); lingxuanze@163.com (X.-Z.L.); zjown0225@163.com (J.Z.); zhangt@yzu.edu.cn (T.Z.); 2College of Animal Science, Shanxi Agricultural University, Taiyuan 030032, China; zhangli030032@163.com (L.Z.); 888lipeifeng888@163.com (P.-F.L.); xmszjc@126.com (Q.-Y.W.)

**Keywords:** miRNA-seq, skeletal muscle, Bian chicken, growth and development

## Abstract

**Simple Summary:**

The development of skeletal muscle during the embryonic period has an important impact on the meat production of poultry, and miRNAs are vital regulators of skeletal muscle development. In this study, through the RNA-seq analysis of leg muscles from 14- and 20-day Bian chicken embryos, we screened out 127 and 131 differentially expressed miRNAs (DEMs) in a fast-growing group (F14 vs. F20) and a slow-growing group (S14 vs. S20), respectively. Based on functional enrichment analysis and interaction network analysis, we further screened out 13 miRNAs and 5 hub genes, which potentially play important roles in skeletal muscle development. This study provides data support for future studies on the regulatory mechanisms of miRNAs in the skeletal muscle development of Bian chickens.

**Abstract:**

MicroRNAs (miRNAs) are widely involved in the growth and development of skeletal muscle through the negative regulation of target genes. In order to screen out the differentially expressed miRNAs (DEMs) associated with skeletal muscle development of Bian chickens at different embryonic ages, we used the leg muscles of fast-growing and slow-growing Bian chickens at the 14th and 20th embryonic ages (F14, F20, S14 and S20) for RNA-seq. A total of 836 known miRNAs were identified, and 121 novel miRNAs were predicted. In the F14 vs. F20 comparison group, 127 DEMs were screened, targeting a total of 2871 genes, with 61 miRNAs significantly upregulated and 66 miRNAs significantly downregulated. In the S14 vs. S20 comparison group, 131 DEMs were screened, targeting a total of 3236 genes, with 60 miRNAs significantly upregulated and 71 miRNAs significantly downregulated. Gene Ontology (GO) and Kyoto Encyclopedia of Genes and Genomes (KEGG) analysis showed that the predicted target genes were significantly enriched in 706 GO terms and 6 KEGG pathways in the F14 vs. F20 group and 677 GO terms and 5 KEGG pathways in the S14 vs. S20 group. According to the interaction network analysis, we screened five coexpressed DEMs (gga-miR-146a-3p, gga-miR-2954, gga-miR-34a-5p, gga-miR-1625-5p and gga-miR-18b-3p) with the highest connectivity degree with predicted target genes between the two comparison groups, and five hub genes (HSPA5, PKM2, Notch1, Notch2 and RBPJ) related to muscle development were obtained as well. Subsequently, we further identified nine DEMs (gga-let-7g-3p, gga-miR-490-3p, gga-miR-6660-3p, gga-miR-12223-5p, novel-miR-327, gga-miR-18a-5p, gga-miR-18b-5p, gga-miR-34a-5p and gga-miR-1677-3p) with a targeting relationship to the hub genes, suggesting that they may play important roles in the muscle development of Bian chickens. This study reveals the miRNA differences in skeletal muscle development between 14- and 20-day embryos of Bian chickens from fast- and slow-growing groups and provides a miRNA database for further studies on the molecular mechanisms of the skeletal muscle development in Bian chickens.

## 1. Introduction

As the most consumed meat in the world, the demand for chicken is on the rise [1]. In recent years, chicken has become the best substitute meat considering the pork shortage, especially affected by African swine fever [2]. In the quest for higher meat production, breeders have carried out chicken breeding since the last century. Many high-yield broiler breeds have been developed to date, such as Avian, Arbor Acres and Ross 308. With the continuous improvement of chicken production, more and more breeders are taking meat quality into account, which has brought indigenous chicken breeds into favor. As an excellent indigenous breed in China, Bian chicken has good meat quality, strong adaptability and is of good value in breeding. Wei et al. [3] demonstrated that the Myf5 and MyoG genes can be used as genetic markers to assist in the selection of superior individuals by analyzing the association between the single-nucleotide polymorphisms (SNPs) of the Myf5 and MyoG genes and growth traits in Bian chickens. He et al. [4] screened differentially expressed genes by sequencing the transcriptome of leg muscles of Bian chickens at 16 weeks of age, providing data support for further study on molecular mechanisms in Bian chicken growth. Apart from research on genes related to Bian chicken growth, there are few studies on miRNA functions in Bian chicken growth.

miRNA is a noncoding sequence with a length of approximately 22 nt, which inhibits the expression of target genes by binding to the 3′UTR region of mRNA [5]. It has been proved to play a pivotal role in many biological processes, including cell proliferation, differentiation and apoptosis [6,7,8]. Numerous studies have demonstrated that some muscle-specific miRNAs highly expressed in muscles, such as miR-1, miR-206, miR-133 and miR-499, can directly or indirectly affect skeletal muscle growth and development [9,10,11]. miR-1 and miR-206 have similar seed sequences and many of the same target genes, such as Pax3, Pax7, HDAC4, Cx43 and Notch3, implying that they play similar regulatory roles in skeletal muscle development [12,13]. Overexpression of miR-1 and miR-206 can promote the differentiation of myoblasts inducing the formation of myotubes [14]. Previous research has produced conflicting results regarding the role of miR-133 in skeletal muscle proliferation and differentiation. Feng et al. [15] demonstrated that miR-133 could inhibit C2C12 cell proliferation but promote differentiation by modulating the extracellular signal-regulated kinase (ERK) signaling pathway. In contrast, Chen et al. [12] found that miR-133 enhances myoblast proliferation by repressing serum response factor (SRF) in C2C12 cells. As far as Horak et al. [5] are concerned, the regulatory role of miR-133 on cell proliferation and differentiation was susceptible to alteration by the surrounding environment. miR-499 was found to be critical for controlling myofiber composition [16]. miR-499 promoted the reprogramming of the fast-to-slow muscle-fiber-type switch and reinforced the exercise endurance capacity by inhibiting the downstream target genes SOX6 and FNIP1 [17]. Although current research on miRNAs involves many fields and species, they are just the tip of the iceberg, and there are still many novel miRNAs waiting for discovery. The application of new technologies such as miRNA chips and high-throughput sequencing has made great contributions to miRNA research.

The number of myofibers in birds has been fixed late in embryonic development, and muscle growth is realized through the hypertrophy of myofibers after birth [18]. Primary myofibers of chickens are formed around the embryonic age of 6, and secondary myofibers begin to differentiate at the embryonic age of 12–16, accounting for a majority of myofibers, which have been basically fixed after the 20-day embryonic age [11,19]. In addition, Gu et al. [20] found the pectoral myofibers of Peking Duck began to change from proliferation to fusion after the 20th embryonic age, accompanied by a decrease in MyoG and MRF4 expression and an increase in MSTN expression. Liu et al. [21] injected IGF-1 into duck embryos and tested them at the 18th, 21th, 24th and 27th embryonic ages. It was found that the weight of duck embryos at the 24th and 27th embryonic ages was significantly higher than that of the control group. Moreover, in ovo injection of IGF-1 also increased the expression levels of MYOG and MRF4, which influenced the differentiation of secondary myofibers by inducing myoblast proliferation. In summary, it is of great importance to improve poultry meat production by studying embryonic muscle development.

Hence, fast- and slow-growing Bian chickens at the 14th and 20th embryonic ages were selected in this study. The differentially expressed miRNAs (DEMs) in two periods were identified by RNA-seq, and their predicted target genes were analyzed for functional enrichment. On this basis, the functions of DEMs were inferred to find miRNAs that potentially affect muscle development in Bian chickens.

## 2. Materials and Methods

### 2.1. Ethics Statement

The experiments involved in this study were completely in compliance with the relevant codes established by the Chinese Ministry of Agriculture. All animal experiments were assessed and approved by the Animal Ethics Committee of Yangzhou University.

### 2.2. Animals and Tissues

The chickens used in this study were Bian chickens, including the fast-growing strain and the slow-growing strain, which were established by Zhang et al. [22]. The bidirectional selection of body weight at 16 weeks was further carried out for 6 generations. At the age of 300 days, 12 female and 1 male Bian chickens in the 7th generation were selected from each group (fast- and slow-growing groups). Each chicken was healthy, and its weight accorded with the group average. In the fast- and slow-growing groups, the average body weight of 12 female chickens was 2248 ± 63.11 g and 1252 ± 24.73 g (mean ± standard deviation), and 2 male chickens weighed 3005 g and 1518 g, respectively. After artificial insemination, the half-sib fertilized eggs from two groups were collected and hatched at 37 °C and 60% humidity.

At the 14th and 20th embryonic ages, eggshells were broken and embryos were rapidly decapitated. Simultaneously, a little blood and allantoic fluid were taken for sexing in order to select female chicken embryos. Subsequently, the collected muscles of left legs were quickly frozen in liquid nitrogen and stored in a refrigerator at −80 °C. The experiment was divided into 4 groups: the 14th embryonic age of the fast-growing group (F14), the 20th embryonic age of the fast-growing group (F20), the 14th embryonic age of the slow-growing group (S14) and the 20th embryonic age of the slow-growing group (S20), with 4 repetitions in each group, totaling 16 chicken embryos.

### 2.3. RNA Extraction and Sequencing

Total RNA was extracted from each leg muscle sample with Trizol (TianGen Biotech, Beijing, China). The qualified RNA was used to construct the cDNA library, which was sequenced on an Illumina 2500 platform after quality inspection.

### 2.4. Quality Control

Q20 and Q30 of the raw reads were calculated to detect the sequencing error rate. High-quality reads without 3′ and 5′ adapter or the insert tag were considered as clean reads. After length selection of clean reads, small RNAs within a certain length range were used for all downstream analyses.

### 2.5. Comparative Analysis and Differential Expression Analysis

Bowtie [23] was used to compare small RNAs meeting the length requirements with the gallus reference sequence to analyze their distribution on the reference sequence. Mapped small RNAs were imported into miRBase for comparison to identify known miRNAs. Novel miRNAs were predicted by miREvo [24] and miRDep2 [25]. The transcripts per million (TPM) algorithm [26] was performed to normalize the expression amount of known and novel miRNAs in each sample. Differential expression analysis of miRNAs from two comparison groups was conducted by the DESeq2 [27] method. The Benjamini and Hochberg method was used for the correction of *p*-values. *p*-adj represents the corrected *p*-value, and *p*-adj < 0.05 was used as the criterion for screening DEMs.

### 2.6. Target Gene Prediction and Enrichment

Target genes of DEMs were predicted by miRanda [28] and RNAhybrid [29], and the intersection of both prediction results was selected.

Gene Ontology (GO) enrichment analysis of target genes was implemented by the GOseq based on Wallenius’ noncentral hypergeometric distribution [30]. Kyoto Encyclopedia of Genes and Genomes (KEGG) enrichment analysis of target genes was performed by KOBAS software [31] to find out the pathways of significant enrichment in target genes.

### 2.7. Interaction Network Analysis

In each group, DEM and target gene pairs were imported into Cytoscape for network analysis, and then the top 10 DEMs were filtered according to the connectivity degree with the predicted target genes. The pathways related to muscle growth and development were selected from the top 20 KEGG pathways, and the enriched genes were imported into the String database for interaction network analysis.

### 2.8. Validation of DEMs by qPCR

Seven DEMs were selected to validate the accuracy of the RNA-seq, including 5 common DEMs in both comparison groups. cDNA synthesis was performed using Mir-X miRNA First-Strand Synthesis Kit (Takara Bio, Beijing, China). Forward primers for all seven miRNAs were designed through miRNA Design V1.01 (Appendix A). Forward and reverse primers of housekeeping gene U6 and universal reverse primers were all from Mir-X miRNA First-Strand Synthesis Kit (Takara Bio, Beijing, China). The melting point of primers was 60 °C. Each primer was verified, and the melting curves had a smooth single peak. Calculation of the relative expression of DEMs was performed using the 2^-ΔΔCT^ algorithm.

## 3. Results

### 3.1. Sequencing Data Analysis

Small RNA (sRNA) transcriptome sequencing was performed on 16 samples of Bian chickens. A total of 202.8 million raw reads were identified in the cDNA library, with an average of 12.67 million raw reads per library, and the Q30 was more than 93%. There were 200.4 million clean reads obtained by quality control. After sRNA length screening, 178.15 million screened reads remained, most of which were concentrated at 21–23 nt in length. This is consistent with the sequence length of miRNA (Figure 1). Compared with the gallus reference sequence, the mapping rate of screened reads was about 97% (Table 1). In addition, there were 836 known miRNAs identified, and 121 novel miRNAs were obtained according to the prediction results of miREvo and miRDeep2.

### 3.2. Differential Expression Analysis of miRNA

In the F14 vs. F20 comparison group, 127 DEMs were screened, with 61 upregulated and 66 downregulated (Figure 2A, Appendix A). In the S14 vs. S20 comparison group, 131 DEMs were screened, with 60 upregulated and 71 downregulated (Figure 2B, Appendix A). In both comparison groups, 91 DEMs were common, indicating that these DEMs play important roles in the development of skeletal muscle of both fast-growing and slow-growing Bian chickens (Figure 3). Cluster analysis showed that the DEMs had different expression patterns in the F14 vs. F20 and S14 vs. S20 groups, and four individuals in each group had excellent reproducibility (Figure 4).

### 3.3. Target Gene Prediction and Functional Enrichment Analysis of DEMs

The intersection of miRanda and RNA hybrid prediction results showed that a total of 127 DEMs targeted 2871 genes in the F14 vs. F20 comparison group (Appendix A), and 131 DEMs targeted 3236 genes in the S14 vs. S20 comparison group (Appendix A). GO and KEGG enrichment analyses were performed on all predicted target genes, in order to reveal the potential role of DEMs in skeletal muscle development. In the F14 vs. F20 comparison group, the target genes of DEMs were significantly enriched in 706 GO terms (*p* < 0.05), including 502 biological process (BP) terms, 73 cellular component (CC) terms and 131 molecular function (MF) terms (Appendix A). For the S14 vs. S20 comparison group, 677 GO terms were significantly enriched (*p* < 0.05), consisting of 475 BP terms, 83 CC terms and 119 MF terms (Appendix A). The top 20 GO terms of each group are shown in Figure 5. Some BP terms related to muscle growth and development were found in both comparison groups, such as mesoderm morphogenesis, positive regulation of RNA biosynthetic process, positive regulation of macromolecule biosynthetic process, multicellular organismal development, etc.

According to KEGG pathway analysis, the target genes of DEMs were significantly enriched in six and five pathways in the F14 vs. F20 and S14 vs. S20 comparison groups (Appendix A), respectively (*p* < 0.05). Other glycan degradation and other types of O-glycan biosynthesis were common significant pathways to both groups. The top 20 KEGG pathways of each group were shown in Figure 6. Some of the pathways associated with muscle growth and development were also listed within the top 20, such as biosynthesis of amino acids, Notch signaling pathway, amino sugar and nucleotide sugar metabolism, protein processing in the endoplasmic reticulum, Hedgehog signaling pathway and TGF-β signaling pathway.

### 3.4. Interaction Network Analysis

According to the connectivity degree with predicted target genes, the top 10 DEMs were screened in the fast-growing and slow-growing groups. There were five DEMs common to both groups (gga-miR-146a-3p, gga-miR-2954, gga-miR-34a-5p, gga-miR-1625-5p, gga-miR-18b-3p). Notably, the connectivity degrees of these five DEMs ranked in the top five of the fast-growing group. In order to further explore the key DEMs and predicted target genes, we performed an interaction network analysis of genes enriched in KEGG pathways associated with muscle growth and development. The interaction network is shown in Figure 7. The analysis showed that a total of 50 genes participated in the construction of the interaction network in the fast-growing group, and the connectivity degree of HSPA5, NOTCH1, PKM2 and NOTCH2 was the highest. For the slow-growing group, the interaction network consisted of 46 genes, with NOTCH1, NOTCH2 and RBPJ having the highest connectivity degree. The prediction results showed that HSPA5 and RBPJ were the targets of gga-let-7g-3p and gga-miR-1677-3p, respectively. PKM2 was the target of gga-miR-490-3p and gga-miR-6660-3p. NOTCH1 was the target of gga-miR-12223-5p in the fast-growing group and novel-miR-327 in the slow-growing group, and NOTCH2 was the target of gga-miR-18a-5p, gga-miR-18b-5p and gga-miR-34a-5p in both groups.

### 3.5. Validation of DEMs by qPCR

Seven DEMs from the F14 vs. F20 and S14 vs. S20 comparison groups were verified for RT-qPCR. The results of qPCR and RNA-seq showed a consistent expression trend, which further demonstrated the accuracy of RNA-seq data (Figure 8).

## 4. Discussion

In this study, high-throughput sequencing technology was used to sequence the miRNAs from fast- and slow-growing Bian chicken muscles at the 14th and 20th embryonic ages. A total of 127 DEMs were screened in the F14 vs. F20 group, and 131 DEMs were screened in the S14 vs. S20 group, including 91 coexpressed DEMs in both groups. The top 10 DEMs of each group were filtered according to the connectivity degree with predicted target genes. Several miRNAs, including gga-miR-146a-3p, gga-miR-2954, gga-miR-34a-5p, gga-miR-1625-5p, and gga-miR-18b-3p, are common to the top 10 DEMs in both groups, and the connectivity degrees of these five miRNAs are ranked in the top 5 of the fast-growing group, suggesting that they play an important role in both fast- and slow-growing Bian chickens, while they may be more significant for fast-growing Bian chickens.

It was reported that miR-146a has a targeting relationship with TGF-β1, which is vital for myoblast differentiation and muscle repair [32,33]. In muscle cells, Smad4 has been proved to be a direct target of miR-146a-5p and is involved in the TGF-β signaling pathway. Downregulation of Smad4 due to miR-146a overexpression would inhibit skeletal muscle fibrosis after acute contusion by interfering with the TGF-β/Smad4 signaling pathway [34]. In our research, the predicted target genes INHBE and AMH of gga-miR-146a-3p were also enriched in the TGF-β signaling pathway, while their function in muscle growth is still unclear. Available studies have shown that miR-2954 is only expressed in birds [35]. Compared to female chicken embryos, Dong et al. [36] found that miR-2954 was more abundantly expressed in male chicken embryos. Moreover, they further confirmed that overexpression of miR-2954 promoted myoblast differentiation, which may be a factor contributing to more myofibers in male chickens. It was reported that the expression level of miR-34a-5p increased with the age of mice in C2C12 cells [37]. Similar results were found in our study, where the expression of gga-miR-34a-5p was significantly higher in Bian chicken leg muscle at the 20th embryonic age compared to the 14th embryonic age in both the fast- and slow-growing groups. In combination with multiple studies [37,38,39], miR-34a functions by inhibiting DNA repair, promoting apoptosis and inducing skeletal muscle aging. The primary regulatory mechanism is considered to decrease the levels of AMP-activated protein kinase by inhibiting the expression of SIRT1, resulting in mitochondrial dynamics dysfunction in skeletal muscle [40]. Numerous studies have proved that miR-18b is a key factor affecting cell differentiation. The expression of miR-18b decreases with higher osteoblast differentiation [41], as the same expression trend was observed in myoblasts in our study. Liu et al. [42] found that TGF-β signal transduction interference due to miR-18b targeting Smad2 would inhibit the differentiation of human hair follicle mesenchymal stem cells into smooth muscle cells. Additionally, it was reported that gga-miR-18b-3p inhibits intramuscular adipocyte differentiation in chickens by downregulating the expression level of ACOT13, revealing its potential role in improving poultry meat quality [43]. Combined with the above views and sequencing results, it can be speculated that gga-miR-18b-3p may also function as a regulator in the differentiation of chicken myoblasts. According to sequencing results, the expression level of gga-miR-1625-5p was significantly downregulated at the 20th embryonic age. Unfortunately, no association with miR-1625-5p has been reported.

Notably, gga-miR-1a-3p was the most abundantly expressed among all DEMs, especially at the 20th embryonic age, with about a 4-fold increase in expression compared to the 14th embryonic age. Wu et al. [44] similarly found extremely abundant expression of gga-miR-1a-3p in 16-week-old Bian chicken leg muscle through miRNA-seq. As one of the muscle-specific miRNAs, miR-1 plays a vital role in the formation of myotubes by regulating the expression of Pax3, HDAC4, YY1, etc. [13,14]. In mice, Wang et al. [45] found that miR-1a-3p targets EIF4E and mediates the Akt/mTOR/S6K pathway to regulate skeletal muscle growth.

All target genes of DEMs were analyzed for KEGG enrichment, and the top 20 KEGG pathways in the fast-growing and slow-growing groups were selected for focus. We found a total of 6 pathways related to muscle growth and development in the top 20 KEGG pathways of the two groups, including biosynthesis of amino acids, the Notch signaling pathway, amino sugar and nucleotide sugar metabolism, protein processing in the endoplasmic reticulum, Hedgehog signaling pathway and TGF-β signaling pathway. A number of key genes such as HSPA5, PKM2, NOTCH1 and NOTCH2 and RBPJ were screened out by performing interaction network analysis of target genes enriched in these pathways. According to the predicted miRNA-mRNA pair, the corresponding miRNAs (gga-let-7g-3p, gga-miR-490-3p, gga-miR-6660-3p, gga-miR-12223-5p, novel-miR-327, gga-miR-18a-5p, gga-miR-18b-5p, gga-miR-34a-5p and gga-miR-1677-3p) were found in further.

It is reported that the endoplasmic reticulum (ER) is essential in the regulation of proteostasis and calcium homeostasis in animal skeletal muscle cells [46]. The overload of misfolded or unfolded proteins in the ER would lead to ER stress [47]. Heat shock protein family A member 5 (HSPA5), as the predicted target gene of gga-let-7g-3p, can initiate the unfolded protein response (UPR) by encoding a binding immunoglobulin protein (BiP) in response to alleviate stress. Accumulating evidence shows that the UPR pathway induced by ER stress plays a pivotal role in the regulation of skeletal muscle mass and metabolic function in multiple conditions. Nakanishi et al. [48] found that the ATF6 transmembrane sensor of the UPR promotes selective apoptosis of myoblasts to eliminate poorly differentiated myoblasts during mouse embryonic development, and further demonstrated that the induction of ER stress has a positive effect on myofiber formation [49]. The PERK/eIF2α signaling axis of the UPR has also been confirmed to be involved in the repair process of skeletal muscle. The expression of eIF2α mutant in mouse skeletal muscle satellite cells could undergo terminal differentiation and subsequent fusion with myofibers [50]. In addition, the UPR pathway is also activated in many cases leading to skeletal muscle atrophy, indicating its regulation of skeletal muscle mass is not always positive [51].

PKM2 is an isoform of pyruvate kinase, which has a pivotal function in glucose metabolism [52]. A large amount of current research on PKM2 is focused on cancer and suggests that PKM2 could be a potential target for cancer therapy. By analyzing the sequencing results, we found that PKM2 was significantly enriched in the KEGG pathway associated with muscle development and was screened as a hub gene in the fast-growing group; hence, we speculate that PKM2 may be associated with the growth and development of Bian chickens. During the lifetime of people without cancer, PKM2 is mainly expressed in the embryonic period and is involved in the repair of tissue [53]. In addition, a study showed that in the skeletal muscle fibers of rats, PKM2 was preferentially expressed in fast-growing myofibers and was expressed in higher abundance than slow-growing myofibers. Overexpression of PKM2 could promote hypertrophy of myotubes [54]. gga-miR-490-3p and gga-miR-6660-3p were predicted to target PKM2. In the miRNA sequencing results of skeletal muscle from MSTN-deficient mutant and wild-type Meishan pigs by Xie et al. [55], miR-490-3p was similarly screened as one of the key miRNAs. On the contrary, no reports have been seen for miR-6660-3p.

The Notch signaling pathway has been proved to regulate myoblast proliferation and differentiation and plays an important role in skeletal muscle regeneration [56,57]. As receptor proteins in the Notch signaling pathway, Notch1 and Notch2 would be cleaved into Notch protein fragments after binding to Notch ligands [58]. These fragments could induce the expression of key genes by transforming the transcription regulatory protein (RBPJ) from repressors to activators in the nucleus [59]. For example, the interaction of activated Notch1 signaling with RBPJ could block the terminal differentiation of muscle progenitors by upregulating Pax7 [60]. Fujimaki et al. [61] found that the proliferation of satellite cell-derived myoblasts was inhibited but premature differentiation was promoted by inactivating Notch1 or Notch2. In the activated state of satellite cells, both Notch1 and Notch2 maintain the proliferative state of satellite cells by blocking differentiation, while the number of satellite cells was also maintained through the cooperation of Notch1 and Notch2 in the quiescent state. In our study, Notch1 was predicted to be a target of gga-miR-12223-5p and novel-miR-327, suggesting that they may be related to skeletal muscle growth and development, although there is no relevant report at present. Notch2 may be regulated by gga-miR-18a-5p, gga-miR-18b-5p and gga-miR-34a-5p according to predictions. It was demonstrated that miR-18a-5p directly targets Notch2 and was involved in regulating the migration and proliferation of smooth muscle cells [62]. miR-18b-5p plays an important role in cancer [63,64], but its function in skeletal muscle is unclear. Although there is no report on miR-34a-5p in chicken skeletal muscle, it has been confirmed to promote apoptosis and senescence in mouse skeletal muscle [37], and miR-34a-5p has the same seed sequence in chickens and mice. As a member of the miR-34 family, miR-34a-5p has a high induction of p53, which induces apoptosis and cell cycle arrest when activated in response to DNA damage or cellular stress [65]. This regulatory effect is positive for suppressing tumorigenesis, while it may play a detrimental role in the repair of skeletal muscle as well.

RBPJ is important for embryos to preserve sufficient satellite cells before birth by preventing the unrestricted differentiation of muscle progenitors [66]. In mouse embryos, the mutation of RBPJ will lead to the overdifferentiation of muscle progenitors and the loss of satellite cells [67]. Therefore, RBPJ is necessary for embryonic development and it will lead to muscular hypoplasia and muscular dystrophy when RBPJ is deficient. Since there is no literature support, whether gga-miR-1677-3p, which was predicted to target RBPJ, plays a regulatory role in skeletal muscle needs further experimental verification.

## 5. Conclusions

Through RNA-seq and bioinformatics analysis of leg muscle for Bian chickens, we screened out five co-expressed DEMs (gga-miR-146a-3p, gga-miR-2954, gga-miR-34a-5p, gga-miR-1625-5p and gga-miR-18b-3p) with the highest connectivity degree with predicted target genes in fast-growing and slow-growing groups. In addition, HSPA5, PKM2, NOTCH1, NOTCH2 and RBPJ were identified as five hub genes according to the further interaction network analysis of genes enriched in KEGG pathways associated with skeletal muscle development. gga-let-7g-3p, gga-miR-490-3p, gga-miR-6660-3p, gga-miR-12223-5p, novel-miR-327, gga-miR-18a-5p, gga-miR-18b-5p, gga-miR-34a-5p and gga-miR-1677-3p were predicted to target these genes, suggesting that they may have a potential regulation in skeletal muscle development. Notably, miR-34a-5p is not only a coexpressed DEM, but also targets one of the hub genes, indicating that it may have a more vital position in Bian chicken growth.

## Figures and Tables

**Figure 1 animals-12-01003-f001:**
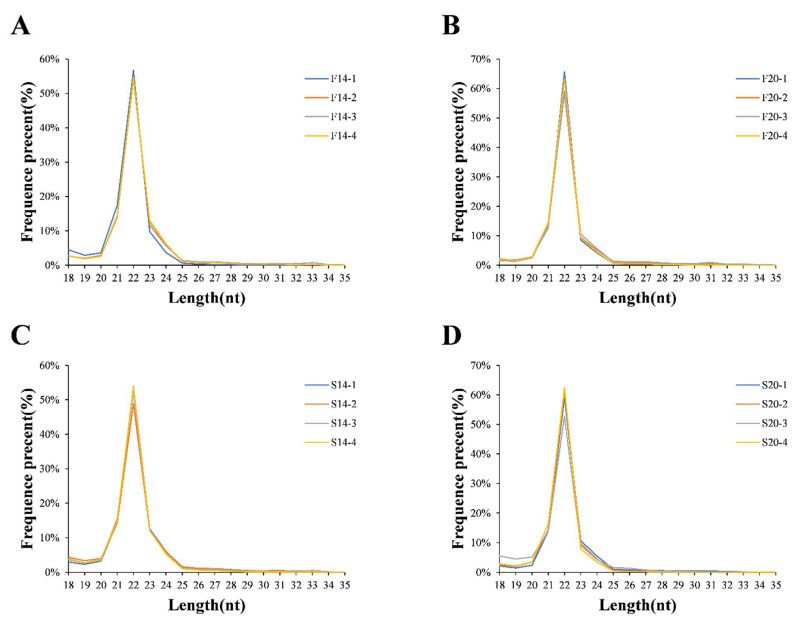
Distribution of length for miRNAs. (**A**) Distribution of length for miRNAs in F14; (**B**) distribution of length for miRNAs in F20; (**C**) distribution of length for miRNAs in S14; (**D**) distribution of length for miRNAs in S20.

**Figure 2 animals-12-01003-f002:**
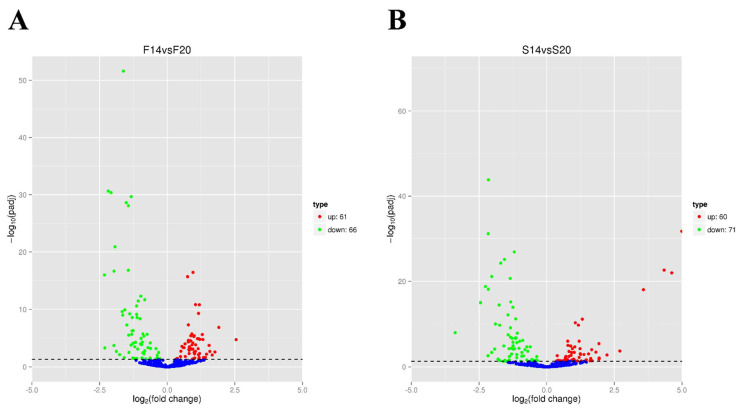
Volcano plot of differentially expressed miRNAs (DEMs). (**A**) Volcano plot of F14 vs. F20 comparison group; (**B**) volcano plot of S14 vs. S20 comparison group.

**Figure 3 animals-12-01003-f003:**
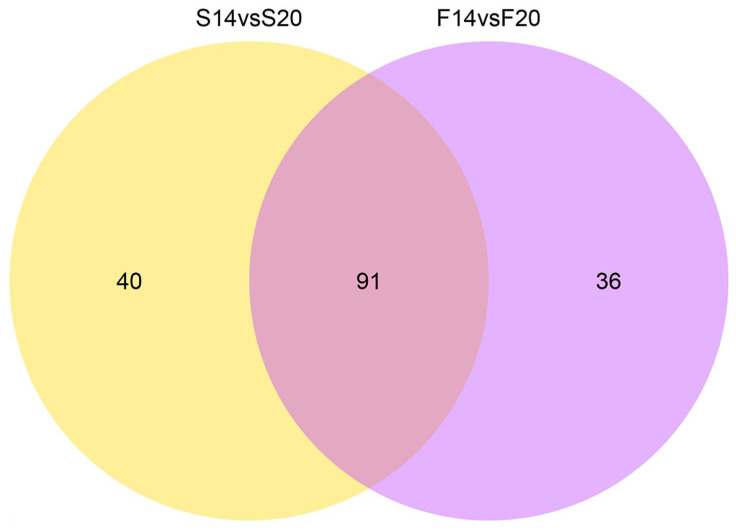
Venn diagram of DEMs in two comparison groups. The circles represent different comparison groups, and numbers represent the number of DEMs.

**Figure 4 animals-12-01003-f004:**
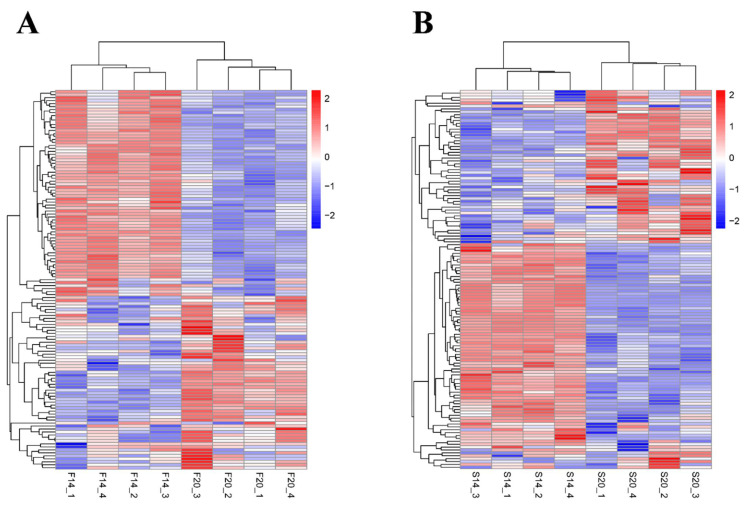
Cluster analysis of DEMs. (**A**) Cluster analysis of F14 vs. F20 comparison group; (**B**) cluster analysis of S14 vs. S20 comparison groups.

**Figure 5 animals-12-01003-f005:**
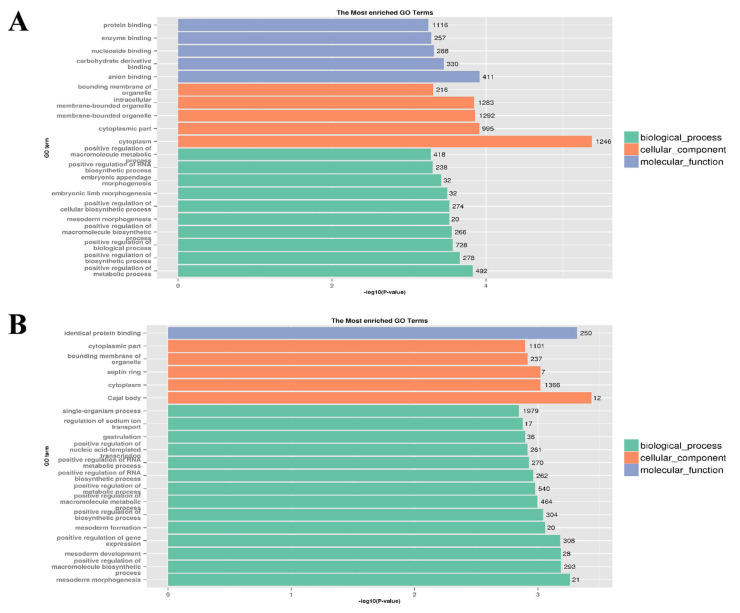
The top 20 GO terms of predicted target genes. (**A**) The top 20 GO terms in F14 vs. F20 comparison group; (**B**) the top 20 GO terms in S14 vs. S20 comparison group.

**Figure 6 animals-12-01003-f006:**
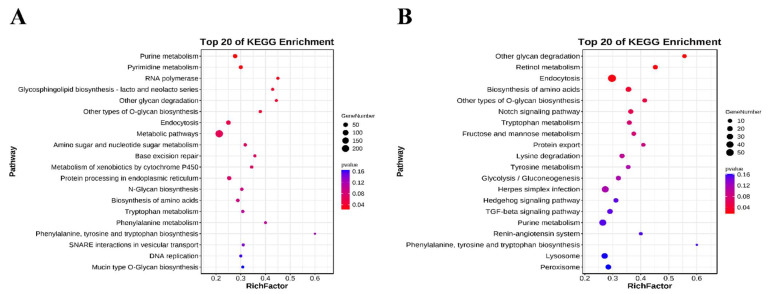
The top 20 KEGG pathways of predicted target genes. (**A**) The top 20 KEGG pathways in F14 vs. F20 comparison group; (**B**) the top 20 KEGG pathways in S14 vs. S20 comparison group.

**Figure 7 animals-12-01003-f007:**
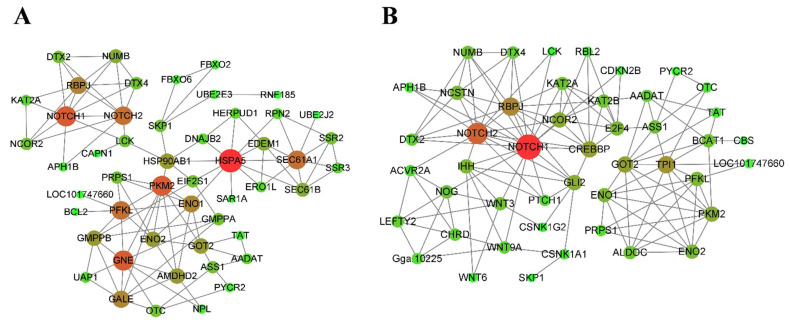
Interaction network analysis. (**A**) Networks of F14 vs. F20 comparison group; (**B**) networks of S14 vs. S20 comparison group.

**Figure 8 animals-12-01003-f008:**
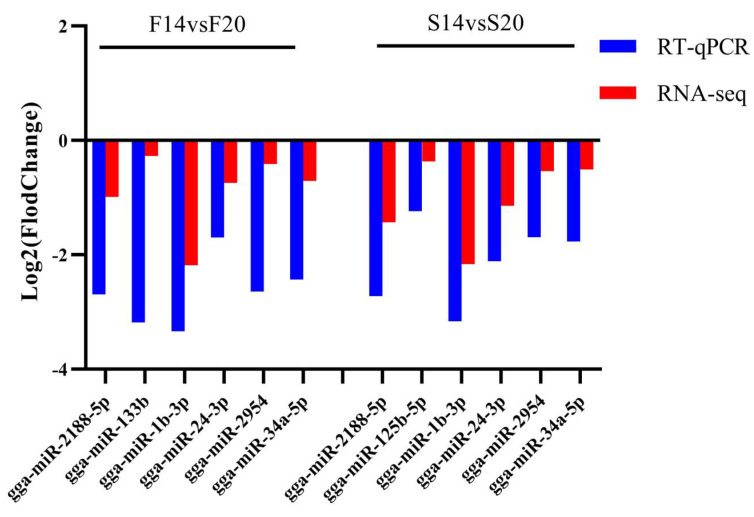
The results of RT-qPCR and RNA-seq for seven DEMs from two comparison groups.

**Table 1 animals-12-01003-t001:** Summary of the filter and mapping of leg muscle miRNA sequencing data.

Sample	Raw Reads	Q30	Clean Reads	Screened Reads	Mapped Reads	Aligned Rate
F14_1	12,669,437	97.19%	12,452,926	10199484	9,992,514	97.97%
F14_2	11,536,200	97.05%	11,441,952	10488628	10,184,164	97.10%
F14_3	11,891,976	97.19%	11,775,156	10588101	10,301,065	97.29%
F14_4	11,314,879	97.03%	11,223,242	10190202	9,906,775	97.22%
F20_1	15,566,534	96.70%	15,393,194	14038143	13,751,156	97.96%
F20_2	14,411,488	96.99%	14,305,379	13522597	13,180,866	97.47%
F20_3	10,756,526	96.71%	10,652,077	10190296	9,926,918	97.42%
F20_4	13,837,711	96.98%	13,682,272	12323470	12,065,830	97.91%
S14_1	12,979,471	97.06%	12,884,152	11,696,696	11,424,755	97.68%
S14_2	14,297,708	97.14%	14,126,008	11,893,624	11,542,015	97.04%
S14_3	11,523,256	97.16%	11,425,892	10,108,473	9,857,892	97.52%
S14_4	14,992,039	93.09%	14,690,516	12,499,530	12,035,961	96.29%
S20_1	12,625,321	97.14%	12,543,865	11,859,676	11,546,364	97.36%
S20_2	11,373,455	95.98%	11,255,753	10,065,046	9,822,986	97.60%
S20_3	10,997,313	97.07%	10,852,319	8,513,819	8,349,604	98.07%
S20_4	12,029,152	92.91%	11,703,244	9,979,970	9,699,893	97.19%

## Data Availability

The data presented in this study are openly available in the Sequence Read Archive (SRA), Reference Number PRJNA773511.

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
