# Peer review of "Comparative Analysis of miRNA Expression Profiles in Skeletal Muscle of Bian Chickens at Different Embryonic Ages"

_animals, 2022, doi:10.3390/ani12081003_

Round 1

Reviewer 1 Report

The manuscript is very interesting and deals with an important and innovative topic in the context of molecular genetics of poultry.
I have a few methodological comments:
- subchapter 2.2 - please provide the number of birds that were intended for this experiment
- subsection 2.3 - from how many individuals the tissues were taken and RNA isolated? was the group representative? How were the collected tissues stored?
provide the manufacturer of the reagent Trizol
- subsection 2.9 - what was the melting point of primers? was the primer optimization performed ??

Author Response

Thank you for your comments and suggestions. I have provided a point-by-point response. Please see the attachment.

Reviewer 2 Report

Dear Authors

Kindly find the attached. I already use (sticky notes) in the PDF file which is a yellow color.

Best regards

Author Response

(The authors gave the same response as above.)

Reviewer 3 Report

The manuscript "Comparative analysis of miRNA expression profiles in skeletal muscle of Bian chickens at different embryonic ages" describes the role of certain microRNAs (miRNA) differentially expressed in skeletal muscles of fast and slow-growing strains of Bian chicken. After determining differentially expressed miRNA at two different embryonic ages, the authors proceed to conduct extensive analysis based on functional enrichment and interaction network analysis for determining which miRNA may play role on skeletal muscle development of Bian chicken. This study is important as it help us understand the mechanics of muscle growth in chickens. 

My main concern is related to the differential of "slow-growing" and "fast-growing" Bian chicken. Besides this basic description by the authors, there is no evidence/fact presented in the paper validating this assertion- no previous publications, no historicity of passaging of the strain, nor evaluation or criteria of evaluation used to deem one strain "fast" and the other "slow". Furthermore, no weights are presented for each individual, nor differential between females and males analysis are presented. There is no statistical comparison of weights in between these two strains at any of the ages tested nor evidence one group would be more "fast growing" upon hatching than the other. 

As the shared PDF did not contained line numbers, I will proceed to comment by page, section, and text:

Page 1. Abstract - In the last sentence, mention that differential analysis has been done between slow- and fast growing strains of Bian chicken. 

Page 2. Introduction- "As the largest consumption meat" - Perhaps "As the largest consumed meat" would be clearer. 

Page 2. Introduction- "become the best substitute with the pork shortage" - Perhaps "become the best substitute meat considering the pork shortage" would be clearer. 

Page 2. 1. Introduction- "Although current researches on miRNAs involves many fields and species," - Perhaps "Although current research on miRNAs involves many fields and species," would be clearer. 

Page 3. 2.Material and Methods; 2.2 Animals and tissues - Provide significant statistical evidence of fast / slow growing strains of Bian chicken used in the study. 

Author Response

(The authors gave the same response as above.)

Round 2

Reviewer 3 Report

The authors have corrected the suggested comments. I have two further comments:

  1. Line 90. Perhaps it should say- "after 20 days of embryonic age" or "after 20-day embryonic age".
  2. Lines 115-116. Variation is included only in females, not in males. It is not specified if the authors are talking about Standard deviation or standard error. Please clarify which dispersion measure is used and add accordingly to the male data as well. 

Author Response

Thank you for your suggestions again. We have made changes in the appropriate places according to your comments.

  1. Line 90. Perhaps it should say- "after 20 days of embryonic age" or "after 20-day embryonic age".

Answer: Thank you for your suggestion. We have changed the original sentence to "after 20-day embryonic age".

  1. Lines 115-116. Variation is included only in females, not in males. It is not specified if the authors are talking about Standard deviation or standard error. Please clarify which dispersion measure is used and add accordingly to the male data as well. 

Answer: We have added “(mean ± standard deviation)” after the female data (Page 3, lines 115-116). In fast-growing and slow-growing strains, 1 male chicken with group average body weight was selected respectively for artificial insemination, and their weight was 3005 g and 1518 g. So there was no dispersion measure in male data.